# Tic Cough in an Adolescent with Organic Brain Pathology—A Case Report and Literature Review

**DOI:** 10.3390/brainsci14010079

**Published:** 2024-01-13

**Authors:** Agnese Grinevica, Agnese Udre, Arturs Balodis, Ilze Strumfa

**Affiliations:** 1Faculty of Medicine, Riga Stradins University, 16 Dzirciema Street, LV-1007 Riga, Latvia; 2Department of Pathology, Riga Stradins University, 16 Dzirciema Street, LV-1007 Riga, Latvia; 3Department of Radiology, Riga Stradins University, 16 Dzirciema Street, LV-1007 Riga, Latvia; 4Institute of Diagnostic Radiology, Paula Stradins Clinical University Hospital, 13 Pilsonu Street, LV-1002 Riga, Latvia

**Keywords:** cough, tic cough, adolescent, hydrocephalus, corpus callosum atrophy

## Abstract

Chronic cough in children and adolescents can be troublesome both to the patient and the whole family. The most common causes of chronic cough in children are protracted bacterial bronchitis and bronchial asthma. However, differential diagnostic workup and treatment can become complicated when a cough of different etiology is encountered, especially in a child having a complex medical history for an unrelated pathology. A cough lacking any identified somatic cause and response to medical treatment in combination with core clinical features of tics that include suppressibility, distractibility, suggestibility, variability, and the presence of a premonitory sensation is labeled tic cough. Here we discuss a case of an adolescent who had atrophy of the corpus callosum and a history of ventriculoperitoneal shunting due to hydrocephalus caused by stenosis of the sylvian aqueduct, but now presented with a debilitating dry cough lasting for several months. After physical causes of cough were ruled out, the diagnosis of tic cough was reached, and multidisciplinary treatment ensured complete recovery. To the best of our knowledge, this is the first reported case showing coincidence of tic cough and hydrocephalus. The co-occurrence of non-syndromic corpus callosum atrophy and tic cough might hypothetically suggest a predisposing pathogenetic link via reduced signaling through cortical inhibitory neurons; further studies are needed. The importance of careful assessment of medical history, clinical picture, and features of the cough itself are emphasized in order to reach the correct diagnosis. Increased awareness of medical society is mandatory to recognize tic cough and to distinguish it from the neurologic manifestations of organic brain pathology.

## 1. Introduction

Cough is one of the most common reasons worldwide why people seek medical help. Despite being a physiologic reflex meant to protect the airways, a cough can be a long-lasting and burdensome condition [1,2]. In children, various viral respiratory infections are considered the most common cause of acute cough, which usually resolves within 4 weeks [3]. In the pediatric population, a cough that lasts more than 4 weeks is defined as a chronic cough [2,3,4]. The most common etiologies of chronic cough in children and adolescents are believed to be protracted bacterial bronchitis and bronchial asthma [3,5].

Psychogenic cough, also referred to as somatic cough syndrome or tic cough, is considered a frequent, but unrecognized cause of chronic cough in children [5,6].

Here we describe a case of a 13-year-old patient with previously diagnosed atrophy of corpus callosum, stenosis of sylvian aqueduct, and ventriculoperitoneal shunt, who now presented with dry, barking–honking cough persisting for several months. Our report highlights the difficulties in diagnosing and managing debilitating chronic cough caused by a transient tic disorder. Increased awareness of healthcare professionals is necessary to recognize the less known causes of chronic cough in children and thus to ensure appropriate and effective treatment.

## 2. Case Presentation

A 13-year-old girl was admitted to the hospital with a severe, debilitating cough, which had started 2 months earlier. The cough was barking; intervals between bouts of cough were approximately 1–2 min. When falling asleep, the cough subsided. There were no other symptoms. The patient and her parents denied headache or any other pain, weight loss or night sweats, visual disturbances, vomiting, and seizures.

The preceding medical history was remarkable for congenital hydrocephalus due to stenosis of the sylvian aqueduct, and atrophy of the corpus callosum. Two years before the current admission, the patient underwent surgical treatment because of neurological symptoms, including severe headache and loss of motor coordination. Endoscopic third ventriculocisternostomy and subsequent ventriculoperitoneal (VP) shunting was performed. After the operation, the symptoms improved significantly, and the headache disappeared. The patient’s intellectual score is in the normal range, and she successfully attends a regular school. Her physical development is appropriate for her age.

Shortly before the onset of the cough in our patient, all family members became ill with a mild viral infection in the upper airways. Everyone but our patient recovered quickly, but she retained a dry, barking cough that interfered with her everyday activities—she could not attend school, meet with friends, go shopping. Her general practitioner subsequently prescribed various medications—inhalations with salbutamol and budesonide, antibacterial therapy with clarithromycin and amoxicillin, and peroral codeine and omeprazole, but those medications did not have any impact on the coughing. Thus, the patient was referred for diagnostics and treatment in the hospital.

On hospital admission, the vital parameters, including body temperature, were within the normal range. The skin over the VP shunt lacked any signs of infection and/or inflammation. Neurological examination showed no abnormalities.

The patient was referred to the pulmonology ward. Various investigations were performed: spirometry showed no evidence of obstructive lung disease; chest X-ray lacked pathological changes; bronchoscopy, including bronchoalveolar lavage, also yielded no pathology. Bacteriological tests, comprising *M. tuberculosis*, were negative. Esophagogastroduodenoscopy showed only mild traits of gastropathy. Laboratory tests were performed, including complete blood count, inflammatory markers such as the C-reactive protein and interleukin-6, as well as basic biochemical tests to assess renal and hepatic function, and levels of glucose and electrolytes. All laboratory tests were within the reference ranges and did not reveal any inflammatory changes or a hypersensitivity reaction. Considering the complete lack of any clinical and laboratory signs of inflammation, as well as absence of neurological signs and symptoms, further workup to rule out a possible VP shunt infection by puncture was not performed. The child was consulted by an otorhinolaryngologist, who excluded any signs of upper airway pathology.

Without finding the explanation for the severe cough, the patient was further investigated towards a central etiology of the cough. Abdominal X-ray did not reveal ventriculoperitoneal shunt catheter migration (Figure 1). Magnetic resonance imaging (MRI) of the brain showed the hydrocephalus with occlusion of the distal part of the aqueductus cerebri and corpus callosum dysplasia with atrophy, consistent with the previous findings. More detailed findings are shown in MRI images (Figure 2).

Consultation of a psychologist was included in the diagnostic workup. Based on the information provided by the mother, the daughter was found to be increasingly protected and cared for. It seemed that the parents helped more than they should, rushed to find positive solutions and consolations for difficult situations, unconsciously preventing their daughter from expressing her concerns, fears, and anxieties that accumulated and appeared in somatization. Consequently, the girl was afraid to experience uncomfortable situations, afraid to experience fear, and also to talk about it. There was a tendency to deny negative feelings and to find positive explanations that brought temporary comfort without solving the inner anxiety. Hypothetically, the illness could be an escape from difficulties such as fear of not keeping up with the learning process, fear of not knowing, and the related anxiety. Consultation also revealed that she had a “strange feeling” in her throat before the coughing episode that resolved after coughing and could be labeled as a premonitory sensation. During consultation, the patient was able to suppress her cough while she was concentrating on some of the activities provided by a specialist.

A child and adolescent psychiatrist established the following diagnoses according to ICD-10: Organic anxiety disorder (F06.4). Transient tic disorder (F95.0). Malformations of sylvian aqueduct (Q03.0). It was recommended to ensure consultations by a psychotherapist, as well as the psychological education of parents. The advised medical therapy comprised a highly selective serotonin reuptake inhibitor, escitalopram 10 mg per day, and an atypical antipsychotic, quetiapine, prescribed in anxiolytic dosage, 25 mg per day. The patient was discharged from the hospital. After three months of treatment, there was no improvement in coughing intensity and frequency, therefore it was decided to switch the medications to second generation antipsychotics. Escitalopram was tapered and aripiprazole was started at dosage of 2.5 mg once per day, titrated to 5 mg once per day. Our patient was slightly sleepy during the first week of therapy, but afterwards tolerated the treatment very well. The patient underwent intensive psychotherapy with a behavioral approach. She also attended a physiotherapy course to increase physical activity with the aim to reduce the anxiety disorder.

During five months of intensive complex treatment, the cough gradually reduced and subsided; treatment with aripiprazole was discontinued. In four months of close surveillance, there was complete remission of the cough and the patient returned to normal life.

## 3. Discussion

Our case report shows a 13-year-old girl who presented to the hospital’s admission department with complaints about a chronic cough lasting for 2 months. The cough was characterized as dry, barking, and progressing. Previously, she had been treated with bronchodilating medications, a proton pump inhibitor, codeine, and antibiotics prescribed by her family physician, with no improvement. As the patient’s condition was troubling and disabling, she was referred to the pulmonology ward for further in-patient investigation.

Her cough lasted more than 4 weeks, thus it was labeled chronic. Although the cough did not respond to medications and did not change its characteristics, it was decided to examine the patient thoroughly to reveal any underlying organic pathology. Protracted bacterial bronchitis and asthma represent the most common causes of chronic cough in children and adolescents [3,5], while gastroesophageal reflux is a controversial cause of chronic cough [7,8].

Prolonged bacterial bronchitis is the most frequent cause of productive chronic cough in preschool children. The median age of such patients ranges from 1.8 years to 4.8 years. However, it could also occur later, but usually not after the age of 12 years. Protracted bacterial bronchitis is diagnosed in 11–41% of children presenting with chronic cough [9]. The gold standard for diagnosis of protracted bacterial bronchitis is flexible bronchoscopy with lavage, but as the method is invasive it is not routinely performed [10]. According to guidelines, supported by Australian clinical trials, it is recommended to establish a diagnosis clinically based on a continuous wet cough of more than four weeks duration, absence of symptoms or signs suggestive of other causes of productive cough, and resolving of symptoms after a two-week course of an appropriate oral antibiotic treatment [9].

Bronchial asthma is another frequent medical problem causing chronic cough, occurring in 14% of children worldwide. The symptoms of asthma include the typical clinical triad of wheezing, shortness of breath, and coughing. Wheezing is considered to be the key feature of asthma and, if not present, a diagnosis of asthma in a child is unlikely [11]. A careful medical history, a clinical examination, and objective tests including spirometry with response to a bronchodilator, fractional exhaled nitric oxide, and prick tests are essential to assess and support the diagnosis [12].

In our case, the most frequent causes of chronic cough could be ruled out by analyzing the characteristics of the cough. Protracted bacterial bronchitis manifests by a chronic productive cough [9,10], but the presented girl had a dry cough. Diagnosis of asthma could be excluded by the constant absence of wheezing which is believed to be a key feature of asthma [11]. Alongside that, her treatment with bronchodilators, prescribed by the family physician, also should be noted both as an indirect hint to another diagnosis as well as an example of overuse of asthma medications. Similarly to this case, a multicenter study on chronic cough in children showed that 70% of children with chronic cough of different origin received asthma medications, although only 15% had a diagnosis of bronchial asthma [13].

As the patient’s condition remained unclear, she underwent several medical investigations, including thoracic X-ray and spirometry, as well as consultation of an otorhinolaryngologist which did not reveal any pathology. Nevertheless, these investigations were important to rule out inflammatory [14,15,16,17] and neoplastic [18,19,20] lung diseases; albeit such conditions (e.g., tuberculosis, hypersensitivity pneumonitis, sarcoidosis, pleuropulmonary blastoma) are rare in children, they still have to be considered in the differential diagnosis. Normal spirometry findings were helpful to further exclude bronchial asthma.

Because of the history of congenital hydrocephalus and ventriculoperitoneal shunting, abdominal X-ray was performed, as dislocation of catheter is reported to cause subdiaphragmatic irritation and cough that undergoes remission after correction of the catheter position [21]. In our patient, there were no radiological data on catheter dislocation. The girl lacked neurological symptoms of shunt dysfunction, including headache, visual disturbances, vomiting, and seizures [22,23]. The shunt infection was excluded, as there were no local or systemic clinical symptoms of infection, and laboratory findings (complete blood count; C-reactive protein) did not indicate inflammation.

Later, an in-depth assessment of her medical history revealed a total absence of coughing during the night. In addition, our patient had a general anxiety disorder (according to ICD-10): anxiety that is generalized and persistent but not restricted to, or even strongly predominating in, any particular environmental circumstances (i.e., it is “free-floating”). The dominant symptoms are variable but include complaints of persistent nervousness, trembling, muscular tensions, sweating, lightheadedness, palpitations, dizziness, and epigastric discomfort. Fears that the patient or a relative will shortly become ill or have an accident are often expressed [24]. Emphasizing the absence of coughing during nighttime, onset of the cough after a viral infection, the coexisting anxiety disorder, the cough being resistant to commonly used medications, and normal findings on medical investigations, a cough of psychogenic origin was suspected [6,25].

Cough occurring in the absence of identified medical disease and lacking response to medical treatment has been labeled as psychogenic cough, habit cough, or tic cough. The proper nomenclature for pediatric nonspecific cough has been the subject of debate in the literature [26]. According to CHEST Guideline and Expert Panel Report guidelines the diagnosis of tic cough can be issued when the patient manifests the core clinical features of tics that include suppressibility, distractibility, suggestibility, variability, and the presence of a premonitory sensation whether the cough is single or one of many tics [6]. On the other hand, psychogenic cough, now referred to as somatic cough syndrome, can only be diagnosed if the patient meets the criteria for a somatic symptom disorder as listed in the Diagnostic and Statistical Manual of Mental Disorders, Fifth Edition [6,26]. This type of cough has been described since 1966. Characteristics of such a cough include dry, repetitive coughing, frequently described as barking or honking [27], sometimes even sounding like a “barking dog” or “barking seal” [28]. Coughing is absent once asleep [6,26,27,28], but can be present repetitively for hours or even during all waking hours. The most common age this disorder occurs is reported to be between age 8 and 13 [27]. Tic cough is frequently combined with other psychiatric and non-psychiatric comorbidities. The most common comorbidity tends to be anxiety disorder. In accordance with this, our patient also was diagnosed with anxiety disorder. The related risk factors in our patient included the parenting style, attentional bias to threat, and perfectionism. In the reported case, anxiety is the main predisposing factor of tic cough. The other risk factors of tics include attention deficit hyperactivity disorder, depression, autism spectrum disorder, learning difficulties, obsessive-compulsive disorder, sleep difficulties, and language difficulties. Our patient did not suffer from any of the mentioned disorders. We cannot exclude that corpus callosum atrophy also might predispose the patient to tics, as discussed further.

Regarding non-psychiatric comorbidities of tic cough, high rates of asthma (22.1%), gastroesophageal reflux (62.3%), and disordered sleep breathing (19.2%) were observed in a recent study [26]. In contrast, to the best of our knowledge, a PubMed search did not yield any case report on tic cough in a patient having hydrocephalus.

As our patient reported that she had a premonitory sensation before coughing, showed throat clearing sounds, and was able to suppress her cough while concentrating on other activities, diagnosis of tic cough, as a part of a transient tic disorder, was established. Considering that cough tic qualifies as a vocal tic, pharmacologic and behavioral therapies shown to be helpful for vocal tics are expected to be useful [6].

After the diagnosis of tic cough was confirmed, we considered the possibility of Tourette’s syndrome s. Tourette’s disorder as it has been associated with chronic persistent cough [29,30], manifests with tics and features anomalies of the corpus callosum that are present in our patient as well.

According to the Diagnostic and Statistical Manual of Mental Disorders, Fifth Ed. (DSM-5), Tourette’s syndrome is characterized by multiple motor tics, and at least one vocal tic, that have persisted for more than a year, although not necessarily simultaneously. To issue the diagnosis of Tourette’s syndrome, toxic effects must be excluded and the symptoms should not be secondary to other diseases. Tourette’s syndrome begins in childhood (before the age of 18 years), classically at the age of 4 to 6 years, and the highest intensity of symptoms is observed at the age of 10 to 12 years [31]. A smaller size of the corpus callosum has been reported in Tourette’s syndrome, performing a cross-sectional case-control study of 158 patients affected by Tourette’s disorder versus 121 healthy controls [32]. Nevertheless, in our case, we excluded this diagnosis due to the isolated occurrence of a single vocal tic, namely, tic cough, not associated with motor tics. However, increased awareness of Tourette’s syndrome would be useful, as the syndrome is notable for its frequent occurrence: it is estimated to affect up to 1% of the population [31].

Our patient’s MRI revealed not only an aqueductal stenosis with subsequent hydrocephalus, but also a corpus callosum dysplasia with atrophy. The corpus callosum is the primary commissural region of the brain consisting of white matter tracts that connect the left and right cerebral hemispheres [33]. A smaller size of the corpus callosum limits neural connections between hemispheres, thus reducing input to cortical inhibitory neurons that are localized in the prefrontal region of the brain [32]. Although greater thickness of the corpus callosum has been reported in individuals with high intelligence quotients [34], even agenesis of the corpus callosum is mostly associated with a quite good prognosis, if this feature is isolated. The affected persons may have difficulties with problem-solving and socializing, placing them in the autism spectrum. Agenesis of the corpus callosum is seen in 3–7 children per 1000 live births and is considered to be sporadic. Hypoplasia of the corpus callosum frequently lacks clinical significance and is identified as an incidental MRI finding when radiological evaluation is carried out for another clinical issue. In turn, dysplasia is a term used to describe an altered morphology (e.g., shape) of the corpus callosum. Atrophic changes, manifesting by thinning of the corpus callosum, can be related to a wide scope of pathologic events, including abnormalities of myelination, hypoxia and ischemia, white matter injuries, a decreased number of neurons, and hydrocephalus (see also Table 1) [35].

Our patient experienced no persistent symptoms that would be unequivocally attributable to pathological changes in the corpus callosum. However, considering the association between vocal tics and a reduced size of the corpus callosum in Tourette’s syndrome [29,30], we cannot exclude that the pathogenesis of tic cough in our patient is associated with atrophic and dysplastic changes in the corpus callosum via reduced signaling in the chains involving cortical inhibitory neurons [32].

In causality studies, analytical statistical comparison of sufficiently large cohorts is considered to be the gold standard to show statistically significant associations between different factors. To consider causality, statistical data should be further supported by experimental studies to detect the sequence and mutual dependence of pathogenetic events. However, authors of an outstanding methodological study [36] have outlined the principles of causality detection, that might be applicable even in the critical assessment of case reports. In particular, three types of horizontal causality and three types of vertical causality have been defined. The horizontal causality represents a relation in which the cause temporally precedes the effect. Vertical causality influences the relation between cause and effect [36]. Evaluating our case by these principles, we reconfirmed our conclusion.

To the best of our knowledge, non-syndromic atrophy of the corpus callosum [37] is not related to immunodeficiency or hypersensitivity that could predispose the patient to protracted bacterial infections or bronchial asthma; these conditions were also excluded in our patient. In contrast, the rare, autosomal recessive Vici syndrome presents with agenesis of the corpus callosum, combined immunodeficiency, developmental delay, cardiomyopathy, congenital cataract, hypopigmentation, and loss of hearing [37]; such manifestations were not present in our patient.

Although the exact mechanism of tic pathophysiology is unknown, the pathogenesis of tics likely involves an abnormality in the central dopaminergic system: excess dopamine in the striatum excites the thalamo-cortical circuits [38,39].

Typical antipsychotics, by blocking D2 receptors and thus inhibiting dopaminergic neurotransmission, have been prescribed to control tic symptoms as first-line agents. Haloperidol and pimozide have historically been used as the first medications for treatment of tic disorder. However, adverse effects, like acute dystonia, parkinsonism, akathisia, and neuroleptic malignant syndrome [40], have led to the use of atypical antipsychotics. Improvement in symptoms has been reported when treating tic cough with atypical antipsychotics, such as risperidone [41,42] and quetiapine [43]. Atypical antipsychotic aripiprazole is one of the alternatives in the treatment of tic disorder. Aripiprazole has a significantly lower adverse effect profile in comparison to both typical and other atypical antipsychotics [44]. The most commonly reported side effects of aripiprazole include restlessness/akathisia, somnolence, and nausea. These effects may be dose-dependent and usually occur early on during treatment, with many patients developing tolerance [45]. Our patient was slightly sleepy during the first week of therapy, but afterwards tolerated the treatment very well. Since aripiprazole affects not only the dopaminergic system, but also serotonergic, GABA-ergic, and glutamatergic systems, it can reduce symptoms of other psychiatric comorbidities, e.g., anxiety, depression, and obsessive–compulsive disorder [46].

To evaluate the therapeutic causality in a case report, a spectrum of criteria has been described [36]. The time-correspondence between treatment and improvement is considered a weak criterion. Strong criteria include time–pattern-correspondence, space–pattern-correspondence, morphological correspondence, dose–effect-correspondence, dialogual correspondence, parallel-test–result-correspondence, and complex-prediction-and-observation correspondence [36]. The last pattern is applicable to our case, along with the dose–effect correspondence for aripiprazole.

Recently, Fujiki et al. studied 151 patients diagnosed with tic cough, and evaluated the effectiveness of behavioral cough suppression therapy. It is a method used by speech–language pathologists and is varied according to the patient’s age. By describing the mechanism of cough, replacement strategies are explained. For example, when there is a premonitory sensation of cough, distraction methods like sipping water, ice cubes, or counting from 1 to 10 can be recommended. In this study, the observed patients experienced symptoms on average 19 months before diagnosis. Behavioral cough suppression therapy reduced coughing in 92.5% of patients following an average of 1.7 sessions [26]. Thus, early behavioral treatment may be a cost-effective, patient-friendly, and efficient option for appropriately diagnosed patients [26,47].

## 4. Conclusions

In conclusion, the differential diagnosis of chronic cough in adolescents and children is wide and not limited to the most frequent causes such as protracted bacterial bronchitis or bronchial asthma. Tic cough represents a less known cause of chronic cough that still must be correctly recognized in order to ensure specific treatment and to reach full recovery while avoiding overtreatment for unrelated diagnoses. To reach this goal, an increased awareness of tic cough is mandatory in the medical society. In unclear cases of chronic cough, evaluation by a child psychiatrist can be useful. Multidisciplinary treatment was involved in this case, including psychotherapy with a behavioral method approach, intensive physiotherapy, and pharmacotherapy with atypical antipsychotics that was monitored by a child and adolescent psychiatrist, leading to complete recovery. The accomplishments of speech therapists should be considered, as behavioral cough suppression therapy is effective in the treatment of tic cough. The coincidence of non-syndromic corpus callosum atrophy and tic cough might hypothetically suggest a predisposing pathogenetic link through reduced signaling via cortical inhibitory neurons; further studies are needed.

## Figures and Tables

**Figure 1 brainsci-14-00079-f001:**
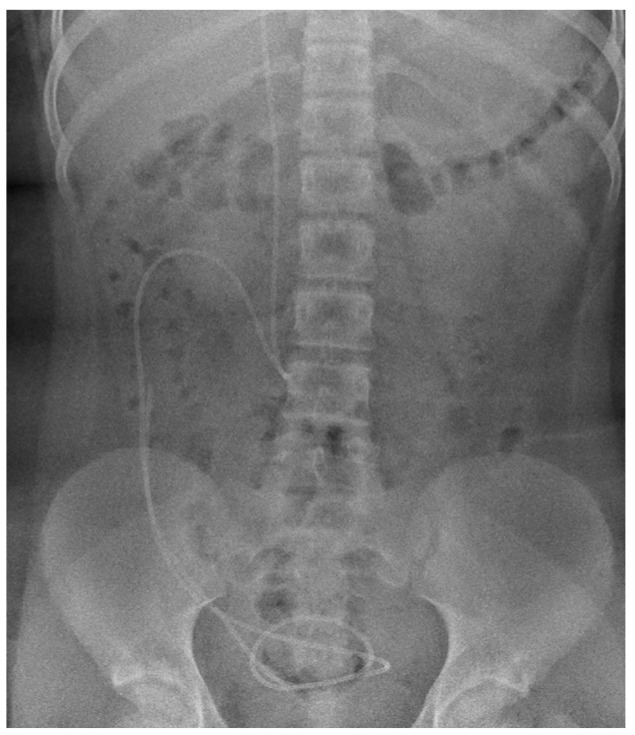
Abdominal X-ray (anteroposterior view) shows the position of the distal part of ventriculoperitoneal shunt catheter, corresponding to the optimal location.

**Figure 2 brainsci-14-00079-f002:**
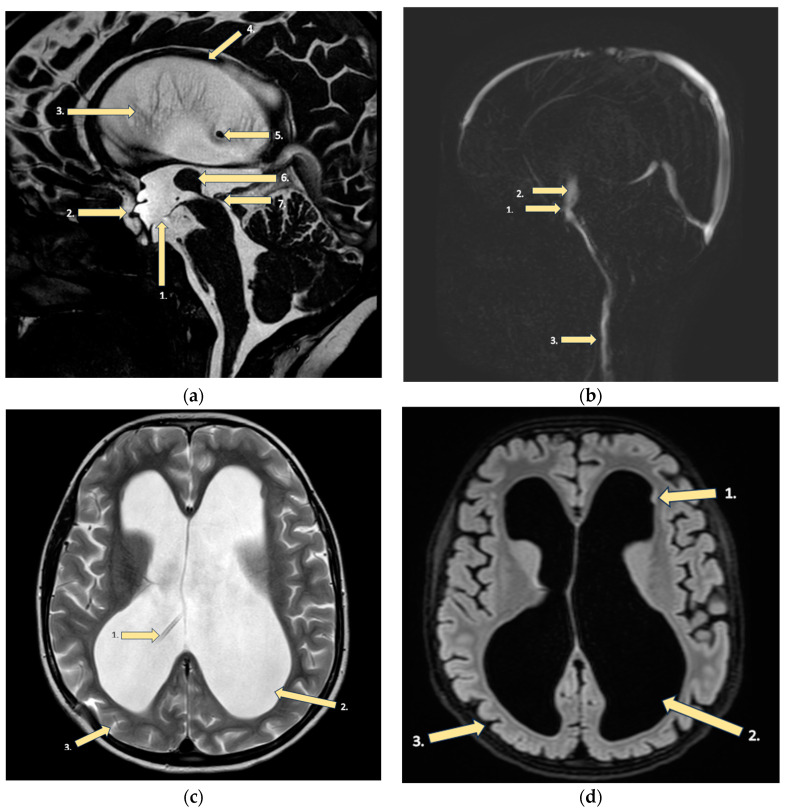
MRI findings of the brain: (**a**) FFE CSF (fast field echo cerebrospinal fluid): (1) Floor of the third ventricle after endoscopic ventriculocisternotomy. (2) Enlargement of the third ventricle’s recesses due to hydrocephalus. (3) Hydrocephalus (dilatation of the lateral ventricles). (4) Corpus callosum dysplasia with underdeveloped genu and body atrophy with periventricular gliosis. (5) End of the ventriculoperitoneal shunt in the right lateral ventricle due to hydrocephalus. (6) Large massa intermedia (large interthalamic adhesion). (7) Occlusion of the distal part of the aqueductus cerebri. (**b**) CSF PCA (principal component analysis): (1) Third ventricle floor with a functioning stoma after ventriculocisternostomy. (2) Increased turbulent flow through ventriculocisternostomy. (3) In the upper segments of the neck intradural flow is synchronous, unhindered. (**c**) T2 sequence: (1) Ventriculoperitoneal shunt in the right lateral ventricle. (2) Hydrocephalus, (3) with reduced brain mass, but without evidence of periventricular edema. (**d**) T2 SPACE dark fluid: (1) Tiny subependymal gray matter heterotopia nodules. (2) Sac-like dilatation of the lateral ventricles (all their parts)—hydrocephalus. (3) Thinner brain parenchyma and minimally enlarged sulci in the brain.

**Table 1 brainsci-14-00079-t001:** Potential causes of atrophic changes in the corpus callosum (based on [35]).

		Type of Atrophy	
	Primary	Secondary	Focal
Mechanism	Primary abnormality of myelination	Diffuse injury	Focal injury
Causes	Hypomyelinating leukoencephalopathies, Metabolic disorders affecting white matter, Microcephaly	Hypoxic–ischemic encephalopathy, HIV-induced encephalopathy, Hydrocephalus, Demyelinating diseases	Dysgenesis of corpus callosum, Localized metabolic disturbances, Hypoglycaemia, Prematurity-related injury of white matter, HIV-related atrophy, Brain infarction and vasculitis, Traumatic brain injury

## Data Availability

The data presented in this study are not publicly available due to containing information that could compromise the privacy of the patient.

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
