# Peer review of "Tic Cough in an Adolescent with Organic Brain Pathology—A Case Report and Literature Review"

_brainsci, 2024, doi:10.3390/brainsci14010079_

Round 1

Reviewer 1 Report

Comments and Suggestions for Authors

The reviewed article shows an interesting case study. It refers to a rare coincidence of tic cough and hydrocephalus in a 13-year-old girl. According to medical records, previously the patient was diagnosed with “stenosis of sylvian aqueduct and ventriculoperitoneal shunt, presenting with dry, barking-honk-ing cough persisting for several months”. After a series of medical examinations and psychological consultations, the patient underwent a 5-month intensive complex treatment. After its completion, subsequent examinations showed complete elimination of cough and an increase in the patient's level of functioning in everyday life. As a reviewer, I believe this article should be published. This is supported by the following considerations: 1. its topic refers to a little-known cause of chronic cough, i.e. tic cough, 2. the authors thoroughly described the diagnostic and therapeutic procedure of the case, which may be helpful in medical practice, 3. the authors point out the need to use multidisciplinary treatment as an approach to increase the effectiveness of treatment in this type of complicated cases.

Author Response

Dear reviewer,

Thank you so much for the careful, detailed analysis of our manuscript and the high estimate of our work. We agree with all your conclusions.

Ilze Strumfa, Professor

Reviewer 2 Report

Comments and Suggestions for Authors

Dear Authors,

I have read a case report regarding tic cough in an adolescent with an organic brain pathology. While this is an interesting case, some major issues need to be resolved:

1) Overall, the authors are not yet linking the cough with organic brain pathology - something that is heavily emphasized in the title. There is a need for a causal relationship or at least an attempt to tie them into one (https://www.ncbi.nlm.nih.gov/pmc/articles/PMC3833527/)

2) Some details are only mentioned in the discussion but not in the case report. For example, the fact that the patient goes to a pulmonologist clinic and the fact that the patient has premonition syndrome before cough need to be mentioned in the case report. 

3) Was there any blood work done?

4) Was there any predisposition towards this mental illness (being tic cough) besides parental concern?

5) The authors need to link back towards the brain pathology. Could the area of the brain affected cause any other causes of cough?

Comments on the Quality of English Language

-

Author Response

Dear reviewer,

Thank you so much for the detailed analysis of our manuscript. We agree with all your comments and have implemented all the recommended corrections:

1) Thank you for an important and interesting remark! We believe that the current body of evidence is insufficient to confirm a causal relationship, but we expanded the discussion in order to comment on the hypothetic pathogenetic link between atrophy of corpus callosum and predisposition to tic cough: "considering the association between vocal tics and reduced size of corpus callosum in Tourette’s syndrome [29,30], we cannot exclude that the pathogenesis of tic cough in our patient is associated with atrophic and dysplastic changes in corpus callosum via reduced signaling from cortical inhibitory neurons [32]" (lines 289 - 293, section 3.Discussion).

2) Thank you for thorough and logical checking of our manuscript. We checked carefully the whole article to ensure that all clinical details are first mentioned in the section 2. Case Presentation. Please, see line 77 for the fact about treatment in pulmonology ward and lines 126 - 129 for the fact that our patient had premonition syndrome.

3) Thank you for the relevant question. Yes, blood tests were performed. We added the following information: "Laboratory tests were performed, including complete blood count, inflammatory markers as the C-reactive protein and interleukin-6, as well as basic biochemical tests to assess renal and hepatic function, levels of glucose and electrolytes. All laboratory tests were within reference ranges and did not reveal any inflammatory changes or a hypersensitivity reaction" (lines 81 - 85; section 2.Case Presentation).

4) Thank you for the remark. We added the comment on risk factors: "The related risk factors in our patient included the parenting style, attentional bias to threat and perfectionism. In the reported case, anxiety is the main predisposing factor of tic cough. The other risk factors of tics include attention deficit hyperactivity disorder, depression, autism spectrum disorder, learning difficulties, obsessive-compulsive disorder, sleep difficulties and language difficulties. Our patient did not suffer from any of the mentioned disorders. We cannot exclude that corpus callosum atrophy also might predispose the patient to tics, as discussed further" (please, see the lines 234 - 240; section 3. Discussion).

5) Once again, thank you for an interesting question! To the best of our knowledge, congenital hydrocephalus and non-syndromic corpus callosum atrophy does not predispose to other causes of cough. We added these consideration in lines 294 - 300 (section 3.Discussion).

6) Minor corrections of English were implemented throughout the article.

Thus, all recommendations are implemented. In the resubmitted manuscript, the changes are highlighted in yellow. On behalf of all authors I would like to thank sincerely for your time input and thoughtful advices that really helped us to improve the manuscript.

Ilze Strumfa, Professor

Reviewer 3 Report

Comments and Suggestions for Authors

Authors present a case report on a 13 years old girl with a severe, debilitating cough, which previously underwent VP shunting for hydrocephalus. This occured following a viral infection where all family members were ill; however, in this patient a protracted cough continued. Workup did not reveal any cause for the cough.  The patients was diagnosed with organic anxiety disorder. After five months, with use of pyschotherapy, behavioral therapy as well as antipyschotics, the cough gradualy reduced.

This manuscript treats an important issue of psychosomatic disorders and tics in younger patients, which are important part of the differential diagnosis, only if all somatic causes were ruled out.

One major issue is that the issue with VP Shunt was not clarified; I suggest to include the preoperative and postoperative imaging. Did the patient receive also third ventriculostomy? Was the x-ray of the shunt performed, was the shunt insufficiency ruled out? Was the shunt infection ruled out (punction of the VP Shunt)? 

Comments on the Quality of English Language

Acceptable. 

Author Response

Dear Reviewer,

Thank you for the detailed analysis of our manuscript. We agree with all your comments and have done our best to provide answers on all the raised questions:

1) Thank you for the question. Yes, the patient received third ventriculostomy. We added this information in lines 61 - 62.

2) Yes, abdominal X-ray was performed to rule out shunt displacement. We added the image (see Figure 1, please). The position of distal part of the shunt was found to be appropriate. Regarding cranial part, the catheter is evident in Figure 2 (image a, arrow N.5). There was no dynamics in MRI images (line 94 - 95). 

3) The shunt insufficiency and shunt infection were ruled out by clinical and laboratory findings. Considering the lack of clinical and laboratory features of inflammation, unfortunately, it was decided not to perform shunt puncture. We added the relevant clinical and laboratory data in lines 56 - 57, 74 - 76 and 81 - 88. We also expanded the discussion as follows: "Because of the history of congenital hydrocephalus and ventriculoperitoneal shunting, abdominal X-ray was made, as dislocation of catheter is reported to cause subdiaphragmatic irritation and cough, that undergoes remission after correction of the catheter position [21]. In our patient, there were no radiological data on catheter dislocation. The girl lacked neurological symptoms of shunt dysfunction, including headache, visual disturbances, vomiting and seizures [22,23]. The shunt infection was excluded, as there were no local or systemic clinical symptoms of infection, and laboratory findings (complete blood count; C-reactive protein) did not indicate inflammation" (lines 198 - 205).

4) Throughout the manuscript, we corrected the level of English language. 

Thus, all the corrections are implemented. In the resubmitted manuscript, the changes are highlighted in yellow. On behalf of all authors, I would like to thank the reviewer for the detailed review and thoughtful comments that really helped us to improve the manuscript. 

On behalf of authors' team,

Ilze Strumfa, Professor

Round 2

Reviewer 2 Report

Comments and Suggestions for Authors

Dear Authors,

Thank you for the revision made. I have only one more point in the case of causality. While I also believe that causality cannot be inferred here, I will refer the authors towards this article (https://pubmed.ncbi.nlm.nih.gov/24416665/), and please add more in-depth discussion about how the underlying suspected cause relate to the outcome of this patient

Comments on the Quality of English Language

-

Author Response

Dear Reviewer,

Thank you so much for raising so interesting question and indicating so brilliant methodological review! We added our considerations to the Discussion (lines 294 - 303 and 331 - 337; highlighted in green). The indicated source is added to References (N.36, highlighted in green). 

Ilze Strumfa

Reviewer 3 Report

Comments and Suggestions for Authors

Authors have sufficiently responded to reviewers remarks. 

Comments on the Quality of English Language

Acceptable. 

Author Response

Dear Reviewer,

Thank you for your time input and the high estimate of our work.

Ilze Strumfa, Professor